# Gross Alpha and Gross Beta Activity Concentrations in the Dust Fractions of Urban Surface-Deposited Sediment in Russian Cities

**Mohamed Y. Hanfi** [1,2,*] **, Ilia Yarmoshenko** [3] **and Andrian A. Seleznev** [1,3]

1   Institute of Physics and Technology, Ural Federal University, Mira St 19, 620002 Ekaterinburg, Russia; sandrian@rambler.ru
2   Nuclear Materials Authority, Maadi, Cairo 520, Egypt
3   Institute of Industrial Ecology Ural Branch of Russian Academy of Sciences, S. Kovalevskoy St. 20, 620219 Ekaterinburg, Russia; ivy@ecko.uran.ru
*   Correspondence: mokhamed.khanfi@urfu.ru

**Abstract:** Studies of gross alpha and gross beta activity in road- and surface-deposited sediments were conducted in three Russian cities in different geographical zones. To perform radiation measurements, new methods were applied which allow dealing with low mass and low volume dust-sized (2–100 μm) samples obtained after the size fractionation procedure. The 2–10 μm fraction size had the highest gross beta activity concentration (GB)—1.32 Bq/g in Nizhny Novgorod and Rostov-On-Don, while the 50–100 μm fraction size was most prominent in Ekaterinburg. This can be attributed to the presence of radionuclides that are transferred through natural and anthropogenic processes. The highest gross alpha activity concentration (GA) in fraction sizes was found in Rostov-on-Don city within the 50–100 μm range—0.22 Bq/g. The fraction sizes 50–100 μm have a higher gross alpha activity concentration than 2–10 μm and 10–50 μm fraction sizes due to natural partitioning of the main minerals constituting the urban surface-deposited sediment (USDS). Observed dependencies reflect the geochemical processes which take place during the formation and transport of urban surface sediments. Developed experimental methods of radiation measurements formed the methodological base of urban geochemical studies.

**Keywords:** gross alpha activity; gross beta activity urban environment; sediment; size fraction





## 1. Introduction

There has been a variety of natural radionuclides in the aquatic and terrestrial ecosystems since Earth's creation. Radionuclides participate in environmental processes such as weathering, sedimentation, resuspension, etc. [1]. Consequently, many studies have measured radionuclide concentrations in various environmental matrices, such as the crust, rocks, sandy beaches, building materials, and the atmosphere [2–4].

Natural radionuclides in minerals and raw materials of natural origin are constantly emitted ionizing radiation that can be exposed to human beings and biota [3,5]. Naturally occurring radioactive materials (NORMs) have resulted from human activities that increase human exposure to Earth's crust radionuclides and can therefore be found in water, air, food, building materials, and the human body [4,6–8]. Radiation hazards are from external and internal exposure to these radioactive isotopes. External exposure is associated with direct gamma radiation emitted from the isotopes in the U and Th series, as well as from $^{40}$K. Internal exposure is caused by the inhalation of inert radioactive gases radon $^{222}$Rn, thoron $^{220}$Rn, and short-lived radioisotopes of their progeny [9,10]. Some artificial radionuclides may be present in the environment (such as $^{137}$Cs and $^{131}$I), such as in Chernobyl [11,12] and due to nuclear weapons testing and nuclear accident. Monitoring of any release of radioactive materials to the environment is necessary for the protection of the environment; for example, if NORM content exceeds the typical background radiation levels, it is

therefore essential to evaluate what precautions should be taken, if any. Additionally, it is suitable to identify the sources of radionuclides, the transportation into the environment, and their migration [13].

In the urban environment, processes such as weathering, soil erosion, as well as anthropogenic impacts on the surfaces produce essential amounts of sediment consisting of grained material of different origin [14,15]. Sediment can deposit in various urban landscape zones and form urban surface-deposited sediment (USDS) which play a significant role in shaping the urban environment [16].

Measuring the gross alpha and gross beta concentration in urban environmental compartments have become increasingly important due to concerns about radioactive environmental contamination through natural and anthropogenic activities resulting in human exposure [17,18]. The objective of the present work is to study the concentration of gross alpha and gross beta activity in size-fractionated samples obtained from the USDS in three Russian cities: Ekaterinburg, Rostov-On-Don, and Nizhny Novgorod. An essential feature of the applied measurements methods is the possibility to detect alpha and beta emitter content in samples of a small amount (mass and volume) of fractionated material.

## 2. Materials and Methods

### 2.1. Description of the Surveyed City

Description of Investigated Cities

The samples of USDS were collected in three Russian cities: Ekaterinburg, Nizhny Novgorod, and Rostov-on-Don [19]. These cities have a continental climate and are located in different geographical zones. The investigated cities are described in Table 1.

**Table 1.** Description of the investigated cities.

| Parameter | Ekaterinburg | Nizhny Novgorod | Rostov-on-Don |
|---|---|---|---|
| Area | 495 km$^2$ | 460 km$^2$ | 348.5 km$^2$ |
| Population | 1,468,833 | 1,259,013 | 1,130,305 |
| Main rivers | Iset | Oka and Volga | Don |
| Latitudes and longitudes | 56°50′ N, 60°35′ E | 56°19 N, 44°00 E | 47°14′ N, 39°42′ E |
| Temperature in July (night/day) °C | 14/24 | 14/24 | 18/29 |
| Temperature in January (night/day) °C | −15/−9 | −11/−5 | −5/−0.1 |
| Climate | Temperate continental | Humid continental | Moderate continental, steppe |
| Geographical region | Eastern slope of the Middle Urals | Valley of the Volga and Oka rivers | Valley of the Don river |
| Geology | Ural Mountains | Alluvial river sediment | Alluvial river sediment |
| Main industries | Productions of machinery, metal processing, metallurgical production, chemical production. | Production of machinery and river shipping | Productions of machinery, river shipping, food industry. |

### 2.2. Sampling Procedure

Approximately 1.5–2 kg of the representative samples of USDS were collected from the surfaces where they were deposited in relatively significant amounts. The samples were put into plastic vacuum bags directly after collection to prevent them from atmospheric moisture. The drying process was carried out under room temperature for one week. Then, two sieving process, decantation and filtration, were performed, which are referred to as wet sieving. Through these processes, the samples were sieved into small-sized fractions which represented dust-sized fractions (2–10 μm, 10–50 μm and 50–100 μm). Dry sieving

for the remainder of the sample was used to fractionate into large-sized elements, which represented fine sand (100–250 μm and 250–1000 μm) and coarse sand (size > 1000 μm) size fractions. The separation by dry and wet sieving is described in Test Method WA 115.1-2017 [20,21].

*2.3. Measurement of Gross Beta Activity*

The method of GB measurements in solid sand and dust samples of low mass (1–10 g) was developed by [11]. For detecting the GB activities, a low background radiometer detector (BDPB-01) was utilized. A plastic scintillation detector with 60 mm diameter and a photomultiplier tube was inserted into a special plastic container. A lead stabilization system of the measuring path was used, which simultaneously enabled testing of the whole path when operating, to promote stability in the disclosure unit. The detection system was shielded by the lead to prevent any external radiation which would impact the beta measurements. The sieved fractions of each sample were weighed and settled in a planchet 2 cm in diameter and 0.6 cm in height. Before the detection of beta in the samples, an empty planchet was assessed for the same counting time using the detector to estimate the background count rate. This process was repeated where the average value of background count rate was 0.017 cpm for beta particles. The GB activity concentration (Bq g$^{-1}$) in the USDS size fractions was computed via the following formula:

$$A_\beta = \frac{I_c - I_{BG}}{\varepsilon(m)\cdot m} \tag{1}$$

where $I_c$ represents the count rate of beta (s$^{-1}$), $I_{BG}$ refers to the background beta count rate (s$^{-1}$), m is the weight of the fractionated sample (g) and the efficiency of detector identified with $\varepsilon(m)$ which depends on m (s$^{-1}$/Bq). The calibration of the detection system is described elsewhere [11].

*2.4. Gross Alpha Measurement Method*

The method of GA measurements in solid grained samples of low mass (about 5 g) was developed by [22]. First, the applied detectors are calibrated using a monazite sample with a known thorium activity concentration (190 ± 15% Bq/g). Twenty-four LR-115 (2.5 × 2.5 cm$^2$) detectors were exposed in direct contact with the monazite sample with a known thorium activity concentration (190 ± 15% Bq/g) for 40 min. After irradiation using the calibration source, the etching process began under standard procedures: a chemical NaOH solution with normality 2.5 N at 50 °C for 2 h [23–25].

After that, a spark counter was employed to register the alpha tracks density in LR-115 films. The calibration factors k, (track cm$^{-2}$ min$^{-1}$/ Bq g$^{-1}$) for the LR-115 films was computed via Equation (2):

$$k = \frac{\rho_t}{A_m\, t} \tag{2}$$

For the GA measurements in the fractionated USDS samples, The LR-115 films (2.5 × 2.5 cm$^2$) were exposed in contact with the fractionated sample (approximately 5 g) and were placed in the hole with a 2 cm diameter for 90 days. During the exposure time, the samples were stored in an accumulation chamber ventilated with fresh air with a low radon concentration where the α particles were released from the radionuclides (238 U, 232 Th and their decay progenies) and formed alpha tracks on the LR-115 film. At the end of exposure time, the LR-115 films were collected and etched under the standard procedures mentioned above. After that, the spark counter was employed to register the alpha track density in LR-115 films. Unexposed LR-115 films were etched and counted via the spark counter to estimate the background alpha track density in the detectors. The GA activity concentration values were estimated by Equation (3) [26]:

$$A = \frac{\rho_t}{k\, t} \tag{3}$$

The uncertainty values were computed for the obtained results and found to be approximately 5% and 3% for GB and GA, respectively. Furthermore, the minimum detectable activity (MDA) values for LR-115 detectors can be computed as follows:

$$\text{MDA} = \frac{\sqrt{N_b} + 2.7}{T\varepsilon} \tag{4}$$

where $N_b$ represents the number of background count rate, T is the exposure duration, and $\varepsilon$ is the detector efficiency. The values of MDA were 0.03 Bq/g, obtained using the Curie standard method [27]. For SSNTDs, the MDA values depend only on the exposure period.

*2.5. Chemical Analysis*

The chemical analysis of the USDS fractionated samples was performed for other studies. The methods of the chemical analyses applied in these studies are described elsewhere [19,20].

The chemical analysis was conducted in the laboratory of the Institute of Industrial Ecology, UB RAS (Ekaterinburg, Russia). Certified methodologies and accreditation by the Russian System of State Accreditation Laboratories of the Institute of Industrial Ecology Chemical Analytical Center provided the quality control for the measurements. The solid fractionated sample was digested utilizing $HNO_3$, $HClO_4$, and HF, pure for analysis [28,29]. Then, the prepared sample solution was analyzed using inductively coupled plasma mass spectrometry (ICP-MS) to detect element concentrations, in particular, U and Th content.

**3. Results**

The descriptive statistics of gross alpha activity concentration (GA), gross beta activity concentration (GB), and uranium and thorium contents in the USDS dust fractions (2–10, 10–50 and 50–100 µm) of Ekaterinburg, Nizhny Novgorod, and Rostov-On-Don are presented in Table 2. As can be seen in Table 2, there is a tendency of variation of radioactive parameters depending on the USDS fractions and the city. The statistical significance of the difference was studied between radioactive parameters for the size fractions in the same city, as well as between the different cities. Due to a low number of measurements of GA which is associated with difficulties of measurements in low-volume samples, the tendencies obtained in GA are insignificant ($p > 0.1$). The dependencies of GB on the size fractions and geographical location are more reliable. The differences between the average GB values in Ekb and RND in size fraction 2–10 µm and 50–100 µm are significant, as is that between Ekb and NN in size fraction 50–100 µm ($p < 0.05$). Analysis of variances confirmed the size fraction and city of sampling as factors influencing the GB ($p < 0.05$).

It is clear that the highest values of GA in the investigated fractions were found in the fraction size 50–100 µm, while the lowest values were observed in the fraction size 2–10 µm for all studied cities. The GB activity concentrations reached the maximum values in the fraction size 50–100 µm for Ekaterinburg, and 2–10 µm for Nizhny Novgorod and Rostov-On-Don. Table 2 presents the chemical compositions obtained in the fraction size; the U and Th content values varied in between various fraction sizes in the investigated cities where the highest U and Th content average values were detected in Ekaterinburg within the fraction size 50–100 µm, and Rostov-On-Don within 10–50 µm, respectively. The minimum average values were recorded in Rostov-On-Don within 50–100 µm and in Ekaterinburg within 2–10 µm, respectively. The distribution of the radioactive parameters is plotted in Figure 1.

**Table 2.** Descriptive statistics for the gross alpha activity concentration (GA), gross beta activity concentration (GB), U content (ppm) and Th content (ppm) in the USDS size fractions (μm).

| City | Descriptive Parameters | GA (Bq g⁻¹) | | | GB (Bq g⁻¹) | | | U (ppm) | | | Th (ppm) | | |
|---|---|---|---|---|---|---|---|---|---|---|---|---|---|
| | | 2–10 | 10–50 | 50–100 | 2–10 | 10–50 | 50–100 | 2–10 | 10–50 | 50–100 | 2–10 | 10–50 | 50–100 |
| Ekaterinburg | Athematic Mean | 0.11 | 0.13 | 0.17 | 0.71 | 0.93 | 1.28 | 1.46 | 2.03 | 2.33 | 4.94 | 4.45 | 4.58 |
| | Geometric mean | 0.1 | 0.12 | 0.16 | 0.61 | 0.67 | 0.93 | 1.22 | 1.48 | 1.66 | 2.14 | 2.74 | 2.67 |
| | SD | 0.06 | 0.02 | 0.04 | 0.43 | 0.86 | 1.13 | 0.80 | 1.40 | 2.05 | 2.30 | 2.34 | 2.30 |
| | Max | 0.18 | 0.15 | 0.20 | 1.72 | 3.20 | 5.30 | 2.90 | 5.16 | 8.26 | 7.02 | 8.65 | 8.11 |
| | Min | 0.06 | 0.11 | 0.12 | 0.28 | 0.15 | 0.20 | 0.31 | 0.08 | 0.17 | 0.14 | 0.10 | 0.10 |
| | *n* | 3 | 4 | 4 | 10 | 23 | 24 | 12 | 14 | 14 | 12 | 14 | 14 |
| Nizhny Novgorod | Athematic Mean | 0.13 | 0.13 | 0.17 | 1.32 | 0.99 | 0.72 | 1.28 | 1.98 | 1.92 | 3.54 | 5.12 | 4.53 |
| | Geometric mean | 0.09 | 0.12 | 0.16 | 0.90 | 0.91 | 0.70 | 1.16 | 1.92 | 1.70 | 2.51 | 4.86 | 4.36 |
| | SD | 0.11 | 0.06 | 0.04 | 1.15 | 0.27 | 0.16 | 0.59 | 0.56 | 1.63 | 2.19 | 1.50 | 1.20 |
| | Max | 0.20 | 0.20 | 0.21 | 4.15 | 1.58 | 1.10 | 2.74 | 3.92 | 10.92 | 9.25 | 7.67 | 7.14 |
| | Min | 0.05 | 0.08 | 0.13 | 0.30 | 0.05 | 0.39 | 0.56 | 1.44 | 1.24 | 1.06 | 2.52 | 2.30 |
| | *n* | 2 | 4 | 3 | 12 | 32 | 35 | 22 | 34 | 34 | 22 | 34 | 35 |
| Rostov On Don | Athematic Mean | 0.15 | 0.19 | 0.22 | 0.95 | 0.90 | 0.69 | 1.52 | 1.94 | 1.97 | 4.64 | 7.45 | 7.35 |
| | Geometric mean | 0.14 | 0.18 | 0.20 | 0.88 | 0.85 | 0.65 | 1.45 | 1.93 | 1.96 | 3.84 | 7.39 | 7.23 |
| | SD | 0.04 | 0.07 | 0.11 | 0.33 | 0.33 | 0.23 | 0.51 | 0.21 | 0.22 | 2.89 | 0.95 | 1.33 |
| | Max | 0.18 | 0.26 | 0.37 | 1.69 | 2.34 | 1.24 | 2.79 | 2.33 | 2.59 | 9.78 | 8.96 | 10.04 |
| | Min | 0.10 | 0.12 | 0.14 | 0.21 | 0.36 | 0.40 | 0.67 | 1.59 | 1.49 | 1.08 | 5.45 | 4.11 |
| | *n* | 3 | 3 | 4 | 31 | 30 | 34 | 17 | 26 | 35 | 17 | 26 | 35 |

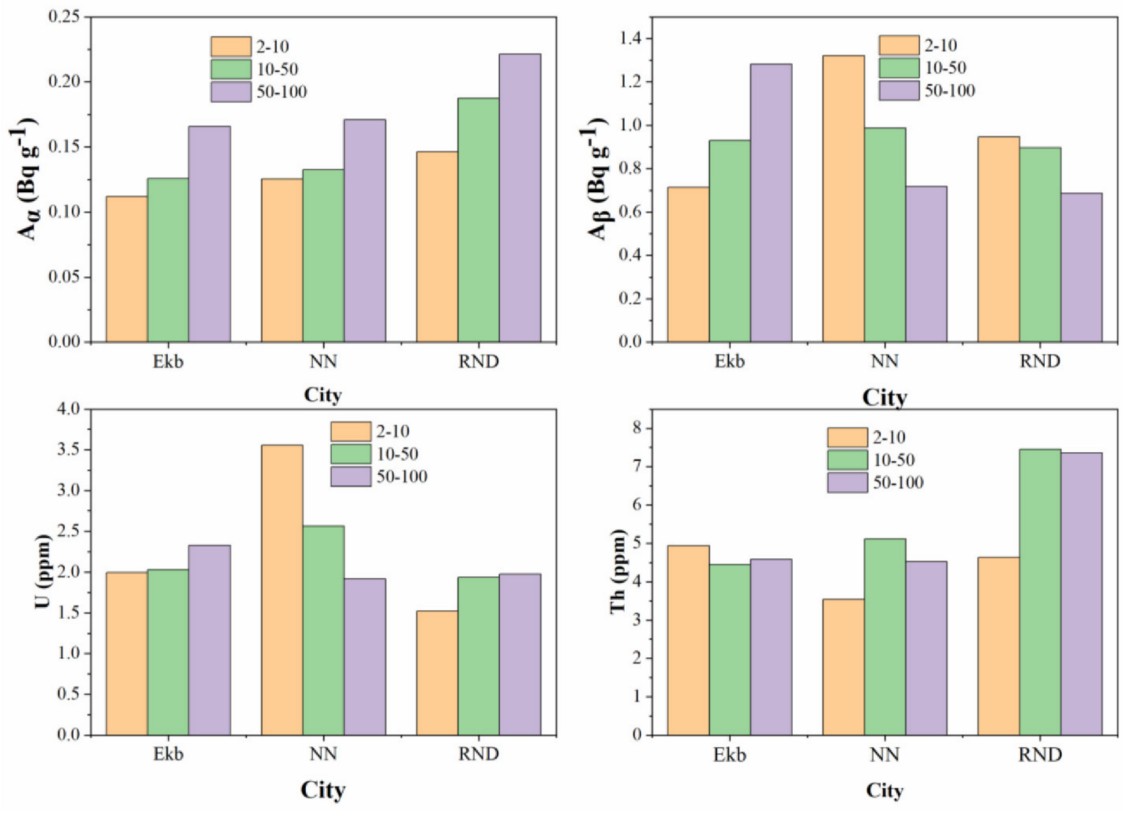

**Figure 1.** The variation of GA and GB within the fraction sizes 2-10, 10-50 and 50-100 μm in the investigated Russian cities.

## 4. Discussion

The detection of GA and GB in the urban environment is an indication of the presence of radionuclides in urban sediments [11]. As clarified from Figure 1, the GA and GB in Ekaterinburg within all fraction sizes had the same natural and anthropogenic origins. Figure 1 illustrates the influence of natural and anthropogenic factors which may be depicted by the results of the GA and GB. The chemical analysis illustrated that USDS contains uranium and thorium, which was higher in the fraction 50–100 μm than in the fractions 2–10 and 10–50 μm in the cities under study. Thus, the GA may be attributed to natural radionuclides in the environment such as uranium, radium, thorium and their decay products [30]. Increasing uranium and thorium content led to increases in the GA activity concentration in the USDS fractions. Moreover, the potassium-40, radium, and decay products were the main beta emitters in the urban sediments. Among the artificial products, agricultural fertilizer, which contains natural radionuclides, led to the increment of potassium (including isotope $^{40}$K) content in USDS fractions [31,32].

The geology of the studied cities can impact the GA and GB in the various fraction sizes. In Ekaterinburg, the geological features are mainly established by the Ural Mountains, while the geologies of Nizhny Novgorod and Rostov-on-Don are related to the alluvial processes of rivers.

Furthermore, the presence of alpha and beta radioactivity can be explained by the migration and transportation of radionuclides from rocks and soils to the urban environment via various pathways such as rainwater, wind, and traffic emissions.

For instance, the correlation between radioactive components of fraction sizes in Ekaterinburg was studied via Pearson's correlation and is presented in Table 3. Strong correlation between GA and GB is obvious, as well between U content (0.99) and Th content (0.74) in the fraction size 2–10 μm. This means the GA and GB are contributed from the same natural and anthropogenic sources. For the fraction size 10–50 μm, the GA and GB were linked with the anthropogenic sources, where the GA changed with GB in opposite directions. The natural sources of fraction sizes 50–100 μm possessed radioactive components; however, the U and Th content changed in opposite direction with the GA.

**Table 3.** Pearson's correlation between radioactive components of fraction sizes in Ekaterinburg.

| 2–10 | GA | GB | Th | U |
|---|---|---|---|---|
| GA | - | | | |
| GB | 1 * | - | | |
| Th | 0.74 | −0.14 | - | |
| U | 0.99 | 0.22 | 0.78 | - |
| 10–50 | GA | GB | Th | U |
| GA | - | | | |
| GB | −0.98 | - | | |
| Th | 0.38 | −0.49 | - | |
| U | 0.31 | −0.38 | 0.90 | - |
| 50–100 | GA | GB | Th | U |
| GA | - | | | |
| GB | −0.60 | - | | |
| Th | −0.93 | 0.06 | - | |
| U | −0.95 | 0.15 | 0.66 | - |

* 3 sample with GA is available.

The main industries in Ekaterinburg are the production of machinery, metal processing, metallurgical production, and chemical production. In Nizhny Novgorod, main industries are the production of machinery and river shipping. Finally, in Rostov-On-Don, productions of machinery, river shipping, and the food industry are dominant.

Domestic emissions, the weathering of facades and pavement surfaces, and the precipitation of previously suspended particles (atmospheric aerosols) are also sources of pollution in residential areas [33–37]. This shows that the GA and GB reflect the migration

and transportation of radionuclides in the urban environment and potentially harmful elements through wind, traffic emissions, and industrial activities from one urban area to others and are closely linked to the examined fraction sizes.

## 5. Conclusions

New methods developed for the measurements of GA and GB in low mass and low volume samples of natural origin were applied to assess the radioactivity of the USDS. The results of the performed measurements were compared with early obtained measurements of the total U and Th concentrations in the same cities. The analysis allows us to draw the following conclusions:

1.  Such natural radionuclides as U, Th, their decay products and $^{40}$K present in the USDS;
2.  Obtained values of GA and GB are generally associated with radionuclides of natural origin. The main sources of natural radioactivity in the urban environment are geological formations and building materials;
3.  Natural radionuclides participate in the sedimentation processes and can be found in the sedimentation material in each city independently of climate, geographical location, and industrial development;
4.  The radioactivity of fine sand and dust fractions can contribute to population radiation exposure in cases of significant resuspension of urban dust by wind and vehicles.

**Author Contributions:** Conceptualization, field sampling, methodology, writing—original draft preparation, M.Y.H., I.Y. and A.A.S.; methodology, writing—review and editing, M.Y.H. and I.Y. All authors have read and agreed to the published version of the manuscript.

**Funding:** The study was supported by Russian Science Foundation (grant No. 18-77-10024).

**Institutional Review Board Statement:** Not applicable.

**Informed Consent Statement:** Not applicable.

**Data Availability Statement:** Not Applicable.

**Conflicts of Interest:** The authors declare no conflict of interest.

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
