# Peer review of "Gross Alpha and Gross Beta Activity Concentrations in the Dust Fractions of Urban Surface-Deposited Sediment in Russian Cities"

_atmosphere, doi:10.3390/atmos12050571_

Round 1

Reviewer 1 Report

The manuscript sounds interesting however I have a few questions and comments. 1. l3 What is "Urban Surface Deposited Sediment" exactly? Is it soil? If it is an atmospheric fallout collected? How such big samples was able to collect if it was an atmospheric fallout? 2. p4 Validation data is needed. 3. l159 What is "fractionated urban sample"? Was it sieved soil? This information in te whole manuscript is misunderstood. In my opinion the idea was it should sound better then "soil". Am I right? 4. l160 "The sample preparation method is similar to the United States Environmental Protection Agency (US EPA) method [28]." How much is it similar? Is it the same or modified? If it is modified it should be written how and what parts. 5. Editorial revision is needed - lack of spaces, different fonts. Some grammar mistakes.

Author Response

Please find attached the submission of the carefully revised version of the manuscript, following the minor comments and modification of the Reviewer.

Below, the detailed list of the changes made in response to the Reviewer’s minor comments (in italics), outlining every change made point by point, is provided. The changes are marked in the manuscript text in red for reviewer #1 and in blue for reviewer #2 .

Reviewer 2 Report

please see attached review.  my main concerns are that:

  1. authors did not identify the source/collection methods/etc. of the dust they had collected from each city.
  2. I am concerned that the differences in total beta/alpha/u/th concentrations in sediments that authors report are not statistically significant given the reported standard deviations.  the authors should identify what differences/trends are STATISTICALLY significant given the sometimes large standard deviations in the data
  3. the conclusions section is simply a list of 4 bulleted points...I am not sure if this is an appropriate use of a conclusions section.
  4.  

Author Response

(The authors gave the same response as above.)

Round 2

Reviewer 1 Report

The manuscript was edited and corrected. There are still some editorial errors that must be corrected however the scientific part looks good. That is why I indicated minor revision. Text must be edited a little bit.

Author Response

                         16 April 2021

Ref: Revision of the manuscript atmosphere-1188291

Title: Gross Alpha and Gross Beta Activity Concentrations in Dust Fraction of Urban Surface Deposited Sediment in Russian Cities

Authors: Mohamed Y. Hanfi, Ilia V. Yarmoshenko and Andrian A. Seleznev

Please find attached the submission of the carefully revised version of the manuscript in Ref., following the minor comments and modification of the Reviewer.

Below, the detailed list of the changes made in response to the Reviewer’s minor comments (in italics), outlining every change made point by point, is provided. The changes are marked in the manuscript text in red for reviewer #1 and in blue for reviewer #2 .

Reviewer #1

Comment: The manuscript was edited and corrected. There are still some editorial errors that must be corrected however the scientific part looks good. That is why I indicated minor revision. Text must be edited a little bit.

Response: Thanks for your recommendations. All the editorial errors are corrected in the revised manuscript. The corrections are marked in the manuscript in green color.

We thank a lot the Reviewer for the useful and valuable comments that have helped to improve the manuscript.

Hoping that all the careful review is sufficient for the direct acceptance of the manuscript, thank you for your time and consideration.

Reviewer 2 Report

I believe the authors have addressed my main concerns and the manuscript can be published after a minor review of grammar and spelling.